# Appearance of Supersonic Stoneley Waves in Auxetics

**Sergey V. Kuznetsov** [1,2]

1 Institute for Problems in Mechanics RAS, 119526 Moscow, Russia; kuzn-sergey@yandex.ru
2 Moscow State University of Civil Engineering, 129337 Moscow, Russia

**Abstract:** It is shown that in auxetic materials (materials with negative Poisson's ratio), supersonic Stoneley waves travelling without attenuation with a velocity equal to or exceeding maximum bulk wave velocity, may exist. Analytical expressions for the relation between negative Poisson's ratio and Young's moduli of the contacting isotropic media ensuring the condition of propagation for supersonic Stoneley waves, are derived by solving a secular equation for Stoneley waves.

**Keywords:** auxetic; Stoneley wave; isotropy; velocity; secular equation; inhomogeneity





## 1. Introduction

It is known [1–9] that genuine Stoneley waves travel along an interface between two dissimilar halfspaces with subsonic velocity, which is smaller than the minimal shear bulk wave velocity in the halfspaces. However, as it is shown theoretically and numerically, the supersonic Stoneley waves may also exist when both halfspaces have a common negative Poisson's ratio (auxetics). Propagation of the interfacial Stoneley waves in linearly elastic homogeneous isotropic and auxetic halfspaces is analyzed by constructing analytical solutions of a secular equation for Stoneley waves [1]. According to [1], the interfacial Stoneley waves propagate along an interface between two acoustically different halfspaces under ideal contact conditions. Thus, the contacting halfspaces form an inhomogeneous space. It is known [1–4] that these waves propagate at a constant speed, independent of frequency; therefore, Stoneley waves do not exhibit dispersion; and in both halfspaces, Stoneley waves exponentially attenuate with depth.

Herein, the secular equation is written in a dimensionless form close to Scholte's secular equation [2,3] for the velocity of Stoneley waves propagating along an interface between two isotropic elastic halfspaces. While Scholte's secular equation has been previously used for deriving the velocity of Stoneley waves, it is being used for the first time for analyzing the velocity of supersonic Stoneley waves, when acoustical properties ensure Stoneley wave velocity exceeding bulk wave velocity in one of the halfspaces.

$$P(c) \equiv c^4 \left( (\rho_1 - \rho_2)^2 - (\rho_1 A_2 + \rho_2 A_1)(\rho_1 B_2 + \rho_2 B_1) \right)$$
$$+ 2Kc^2(\rho_1 A_2 B_2 - \rho_2 A_1 B_1 - \rho_1 + \rho_2) + K^2(A_1 B_1 - 1)(A_2 B_2 - 1) = 0 \quad (1)$$

where $c$ is the Stoneley wave velocity; $\rho_k$, $k = 1, 2$ are material densities of the contacting halfspaces, and

$$K = 2\left( \rho_1 \beta_1^2 - \rho_2 \beta_2^2 \right), \quad A_k = \sqrt{1 - \frac{c^2}{\alpha_k^2}}, \quad B_k = \sqrt{1 - \frac{c^2}{\beta_k^2}}, \quad k = 1, 2 \quad (2)$$

In Equation (2) $\alpha_k$, $\beta_k$, $k = 1, 2$ are longitudinal and shear bulk wave velocities:

$$\alpha_k = \sqrt{\frac{\lambda_k + 2\mu_k}{\rho_k}}, \quad \beta_k = \sqrt{\frac{\mu_k}{\rho_k}} \quad (3)$$

where $\lambda_k$, $\mu_k$, $k = 1, 2$ are Lame's constants.

The majority of theoretical works on Stoneley wave analyses [1–9] are concerned with the case of subsonic velocities, satisfying condition

$$c < \min_{k=1,2}(\beta_k) \tag{4}$$

Condition Equation (4) implies all the coefficients in the Stoneley equation to be real, and, as was asserted in [1], it is necessary for the existence of a positive root of the secular Equation (1) associated with the Stoneley wave velocity. Other methods for analyzing Stoneley waves propagating with subsonic velocities obeying condition Equation (4) relate to different variants of sextic formalisms, Stroh [10,11] and Cauchy [12], along with various asymptotic techniques [13–15]. The principal ability for leaky Stoneley waves propagating with velocities exceeding the limit in the right-hand side of Equation (4) is asserted in [10,11].

Herein, an explicit analytical condition between physical properties of the auxetic media is derived, ensuring the existence of supersonic leaky Stoneley waves propagating with velocity

$$c = \max_{k=1,2}(\alpha_k). \tag{5}$$

The derivation is based on constructing analytical solution for Stoneley secular Equation (1) at a special condition

$$\rho_1\beta_1^2 = \rho_2\beta_2^2 \tag{6}$$

Condition Equation (6) is in some cases less restrictive than the Wiechert condition [16,17] frequently used at Stoneley wave analyses [1–8]:

$$\frac{\lambda_1}{\lambda_2} = \frac{\mu_1}{\mu_2} = \frac{\rho_1}{\rho_2} \tag{7}$$

since in view of Equation (3), condition Equation (6) imposes restriction on $\mu_k$, $k = 1, 2$ only, demanding

$$\mu_1 = \mu_2. \tag{8}$$

Note that condition Equation (8) does not necessarily imply $\lambda_1 = \lambda_2$. Actually, condition Equation (6) resembles the condition for equal acoustic impedances [17]. However, as will be shown below, condition Equation (6) is much less restrictive than the condition of equal acoustic impedances.

## 2. Solving Secular Equation

### 2.1. Basic Relations

Consider stratified inhomogeneous space consisting of two isotropic and homogeneous halfspaces with ideal mechanical contact between them. Introducing dimensionless Lame's constants $\widetilde{\lambda}$, $\widetilde{\mu}$ and dimensionless density $\widetilde{\rho}$

$$\widetilde{\lambda} = \frac{\lambda_1}{\lambda_2}, \quad \widetilde{\mu} = \frac{\mu_1}{\mu_2}, \quad \widetilde{\rho} = \frac{\rho_1}{\rho_2} \tag{9}$$

the dimensionless bulk wave velocities in the halfspaces take the form

$$\widetilde{\alpha} = \frac{\alpha_1}{\alpha_2}, \quad \widetilde{\beta} = \frac{\beta_1}{\beta_2} \tag{10}$$

with notations Equations (9) and (10), secular Equation (1) can be rewritten in terms of the dimensionless variables

$$P(\widetilde{c}) \equiv L\widetilde{c}^4 + 2M\widetilde{c}^2 + N = 0 \tag{11}$$

where $\widetilde{c}$ is the dimensionless velocity

$$\widetilde{c} = \frac{c}{\beta_2} \tag{12}$$

and

$$\begin{aligned}
L &= \left( (\widetilde{\rho} - 1)^2 - \left( \widetilde{\rho}\widetilde{A}_2 + \widetilde{A}_1 \right)\left( \widetilde{\rho}\widetilde{B}_2 + \widetilde{B}_1 \right) \right) \\
M &= \widetilde{K}\left( \widetilde{\rho}\widetilde{A}_2\widetilde{B}_2 - \widetilde{A}_1\widetilde{B}_1 - \widetilde{\rho} + 1 \right) \\
N &= \widetilde{K}^2\left( \widetilde{A}_1\widetilde{B}_1 - 1 \right)\left( \widetilde{A}_2\widetilde{B}_2 - 1 \right)
\end{aligned} \tag{13}$$

In Equation (13) coefficient $\widetilde{K}$ has the form

$$\widetilde{K} = 2\left( \widetilde{\rho}\widetilde{\beta}^2 - 1 \right) \tag{14}$$

and

$$\widetilde{A}_k = \sqrt{1 - \widetilde{c}^2\frac{\beta_2^2}{\alpha_k^2}}, \qquad \widetilde{B}_k = \sqrt{1 - \widetilde{c}^2\frac{\beta_2^2}{\beta_k^2}}, \qquad k = 1, 2 \tag{15}$$

### 2.2. Secular Equation at Condition

In terms of the dimensionless parameters condition Equation (6) reads as

$$\widetilde{\rho}\widetilde{\beta}^2 = 1 \tag{16}$$

ensuring

$$\widetilde{K} = 0 \tag{17}$$

The latter yields

$$M = N = 0 \tag{18}$$

Now, at Equation (18), Equation (11) becomes

$$L\widetilde{c}^4 = 0 \tag{19}$$

Equation (19) may have nontrivial solution(s) with respect to $\widetilde{c}$, if

$$L = 0 \tag{20}$$

Suppose now that both contacting media have equal Poisson's ratios then in view of condition Equation (16), relative bulk velocities become

$$\widetilde{\alpha} = \widetilde{\beta} = \frac{1}{\sqrt{\widetilde{\rho}}} \tag{21}$$

and

$$\begin{aligned}
\widetilde{A}_1 &= \sqrt{1 - \frac{\widetilde{c}^2}{\widetilde{\rho}\gamma^2}}, \qquad \widetilde{A}_2 = \sqrt{1 - \frac{\widetilde{c}^2}{\gamma^2}} \\
\widetilde{B}_1 &= \sqrt{1 - \frac{\widetilde{c}^2}{\widetilde{\rho}}}, \qquad \widetilde{B}_2 = \sqrt{1 - \widetilde{c}^2}
\end{aligned} \tag{22}$$

where

$$\gamma = \sqrt{2}\sqrt{\frac{1 - \nu}{1 - 2\nu}} \tag{23}$$

In Equation (23) $\nu$ is the common Poisson's ratio of both media.

## 3. Supersonic Stoneley Waves

Consider now the case when velocity of Stoneley waves satisfies supersonic condition Equation (5), then Equation (20) with account of Equations (21) and (22), yields

$$
\begin{cases}
(\widetilde{\rho}-1)^2 - \widetilde{\rho}\sqrt{1-\widetilde{\rho}}\left(\widetilde{\rho}\sqrt{1-\gamma^2}+\sqrt{1-\widetilde{\rho}\gamma^2}\right)=0, & \widetilde{\rho}>1 \;\;\&\;\; \widetilde{c}^2=\widetilde{\rho}\gamma^2 \\
(\widetilde{\rho}-1)^2 - \sqrt{1-\frac{1}{\widetilde{\rho}}}\left(\widetilde{\rho}\sqrt{1-\frac{\gamma^2}{\widetilde{\rho}}}+\sqrt{1-\gamma^2}\right)=0, & \widetilde{\rho}<1 \;\;\&\;\; \widetilde{c}^2=\gamma^2
\end{cases}
\tag{24}
$$

Note that parameter $\gamma \in \left(\frac{2}{\sqrt{3}};\ \infty\right)$ at $\nu \in (-1,\ 0.5)$, thus the left-hand sides of Equation (24) are real, despite all imaginary radicands.

### 3.1. Solutions for γ

Equation (24) admit analytical closed form solutions in terms of relations between $\widetilde{\rho}$ and $\gamma$:

$$
\gamma = \begin{cases}
\widetilde{\rho}^{-3/2}\sqrt{(\widetilde{\rho}+1)\left(\widetilde{\rho}^2+\widetilde{\rho}-1\right)-2\sqrt{\widetilde{\rho}(\widetilde{\rho}-1)\left(\widetilde{\rho}^2+\widetilde{\rho}-1\right)}} \\[2mm]
\widetilde{\rho}^{-3/2}\sqrt{(\widetilde{\rho}+1)\left(\widetilde{\rho}^2+\widetilde{\rho}-1\right)+2\sqrt{\widetilde{\rho}(\widetilde{\rho}-1)\left(\widetilde{\rho}^2+\widetilde{\rho}-1\right)}} \\[2mm]
\qquad \widetilde{\rho}>1 \;\;\&\;\; \widetilde{c}^2=\widetilde{\rho}\gamma^2
\end{cases}
\tag{25}
$$

and

$$
\gamma = \begin{cases}
\sqrt{(\widetilde{\rho}+1)\left(1-\widetilde{\rho}^2+\widetilde{\rho}\right)-2\widetilde{\rho}\sqrt{(1-\widetilde{\rho})\left(1-\widetilde{\rho}^2+\widetilde{\rho}\right)}} \\[2mm]
\sqrt{(\widetilde{\rho}+1)\left(1-\widetilde{\rho}^2+\widetilde{\rho}\right)+2\widetilde{\rho}\sqrt{(1-\widetilde{\rho})\left(1-\widetilde{\rho}^2+\widetilde{\rho}\right)}} \\[2mm]
\qquad \widetilde{\rho}<1 \;\;\&\;\; \widetilde{c}^2=\gamma^2
\end{cases}
\tag{26}
$$

### 3.2. Solutions for ν

Relations Equation (25), can also be expressed in terms Poisson's ratio and $\widetilde{\rho}$ by use of Equation (23):

$$
\nu = \begin{cases}
\frac{1}{2}\dfrac{\widetilde{\rho}^3-2\widetilde{\rho}^2+1+2\sqrt{\widetilde{\rho}(\widetilde{\rho}-1)\left(\widetilde{\rho}^2+\widetilde{\rho}-1\right)}}{1-2\widetilde{\rho}^2+2\sqrt{\widetilde{\rho}(\widetilde{\rho}-1)\left(\widetilde{\rho}^2+\widetilde{\rho}-1\right)}} \\[2mm]
\widetilde{\rho}>1 \;\;\&\;\; \widetilde{c}^2=\widetilde{\rho}\gamma^2
\end{cases}
\tag{27a}
$$

$$
\nu = \begin{cases}
\frac{1}{2}\dfrac{\widetilde{\rho}^3-2\widetilde{\rho}^2+1-2\sqrt{\widetilde{\rho}(\widetilde{\rho}-1)\left(\widetilde{\rho}^2+\widetilde{\rho}-1\right)}}{1-2\widetilde{\rho}^2-2\sqrt{\widetilde{\rho}(\widetilde{\rho}-1)\left(\widetilde{\rho}^2+\widetilde{\rho}-1\right)}} \\[2mm]
\widetilde{\rho}>1 \;\&\; \widetilde{c}^2=\widetilde{\rho}\gamma^2
\end{cases}
\tag{27b}
$$

and

$$
\nu = \begin{cases}
\frac{1}{2}\dfrac{\widetilde{\rho}^3-2\widetilde{\rho}+1+2\widetilde{\rho}\sqrt{(1-\widetilde{\rho})\left(1+\widetilde{\rho}-\widetilde{\rho}^2\right)}}{\widetilde{\rho}^2-2+2\sqrt{(1-\widetilde{\rho})\left(1+\widetilde{\rho}-\widetilde{\rho}^2\right)}} \\[2mm]
\widetilde{\rho}<1 \;\;\&\;\; \widetilde{c}^2=\gamma^2
\end{cases}
\tag{28a}
$$

$$
\nu = \begin{cases}
\frac{1}{2}\dfrac{\widetilde{\rho}^3-2\widetilde{\rho}+1-2\widetilde{\rho}\sqrt{(1-\widetilde{\rho})\left(1+\widetilde{\rho}-\widetilde{\rho}^2\right)}}{\widetilde{\rho}^2-2-2\sqrt{(1-\widetilde{\rho})\left(1+\widetilde{\rho}-\widetilde{\rho}^2\right)}} \\[2mm]
\widetilde{\rho}<1 \;\;\&\;\; \widetilde{c}^2=\gamma^2
\end{cases}
\tag{28b}
$$

### 3.3. Numerical Analysis

All four solutions in Equations (27) and (28) lead to auxetics (materials with negative Poisson's ratio [18]), in particular, solution (27a) leads to negative Poisson's ratios at $\widetilde{\rho} \in (1;\ \sim 1.23)$; solution (27b) leads to negative Poisson's ratios at $\widetilde{\rho} \in (\sim 3.8;\ \sim 12.0)$; solution (28a) leads to negative Poisson's ratios at $\widetilde{\rho} \in (\sim 0.82;\ 1)$; and solution (28b) yields negative Poisson's ratios at $\widetilde{\rho} \in (\sim 0.09;\ 0.25)$.

The plots revealing the variation of Poisson's ratio vs relative material density $\widetilde{\rho}$ corresponding to solutions (27a) and (27b) are shown in Figure 1.

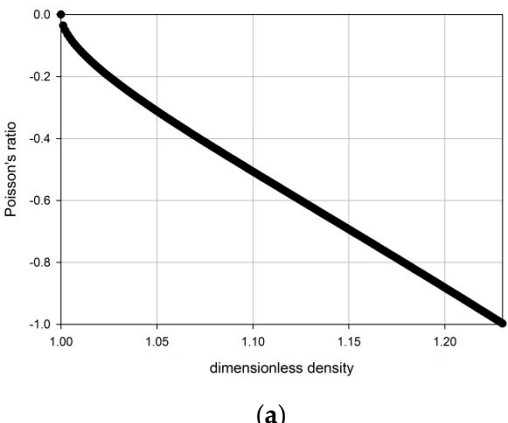

(**a**)

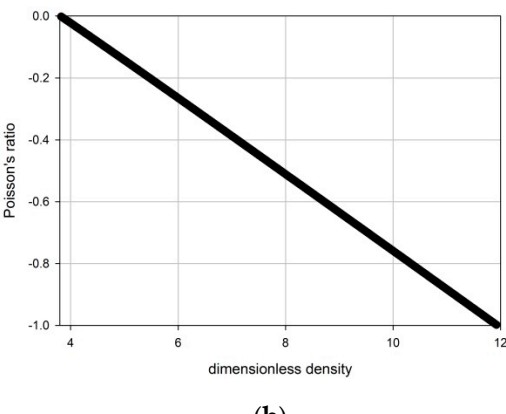

(**b**)

**Figure 1.** Variation of Poisson's ratio vs. dimensionless density in auxetic media: (**a**) Equation (27a); (**b**) Equation (27b).

The plots obtained by Equations (27a) and (27b) are shown in Figure 2.

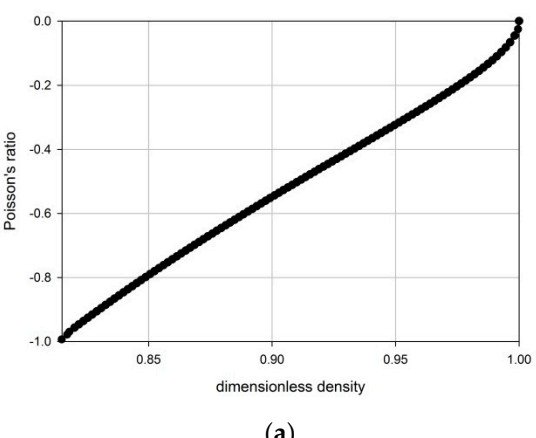

(**a**)

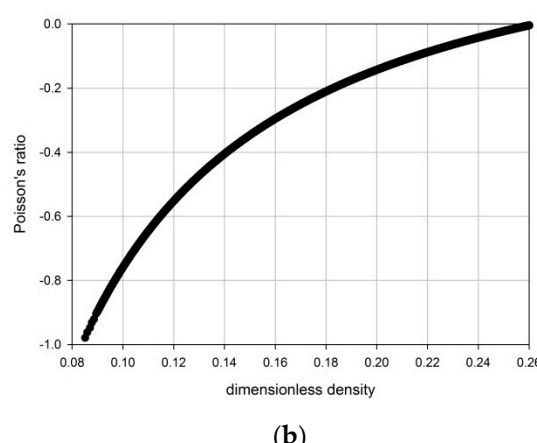

(**b**)

**Figure 2.** Variation of Poisson's ratio vs dimensionless density in auxetic media: (**a**) Equation (28a); (**b**) Equation (28b).

Thus, at both $\widetilde{\rho} < 1$ and $\widetilde{\rho} > 1$ there are intervals of relative material densities admitting propagation of non-attenuating leaky Stoneley waves for the considered auxetic media with common negative Poisson's ratios; in this respect, see also [19–21]. It should also be noted that according to [22], auxetic materials need not be necessary anisotropic, but they may belong to the isotropic class. Another remark concerns some of the related works on acoustic wave propagation in media with more complicated properties; in this respect, see [23] for nonlinear waves and [24] for Stoneley waves in media satisfying the Wiechert condition.

## 4. Concluding Remarks

The obtained analytical expressions Equations (27) and (28) for the common Poisson's ratios of the auxetic media and their densities, define supersonic leaky Stoneley waves propagating without attenuation in the direction of propagation along the interface plane, due to real values for the Stoneley wave velocity.

In contrast to the non-attenuating (in the direction of propagation) leaky Rayleigh waves that arise only at special kinds of elastic anisotropy [10,11], the analyzed supersonic and non-attenuating leaky Stoneley waves propagating with velocities

$$\begin{cases} \widetilde{c}^2 = \widetilde{\rho}\gamma^2 & @ \quad \widetilde{\rho} > 1 \\ \widetilde{c}^2 = \gamma^2 & @ \quad \widetilde{\rho} < 1 \end{cases} \tag{29}$$

exist at the contacting isotropic auxetic halfspaces, satisfying condition Equation (16). Also note that at the intermediate velocity range

$$\begin{cases} \widetilde{\beta} < \widetilde{c}^2 < \widetilde{\rho}\gamma^2 & @ \quad \widetilde{\rho} > 1 \\ 1 < \widetilde{c}^2 < \gamma^2 & @ \quad \widetilde{\rho} < 1 \end{cases} \tag{30}$$

non-attenuating in direction of propagation leaky Stoneley waves cannot exist because of appearing imaginary terms in secular Equation (24), leading to complex values for velocity $\widetilde{c}$, and hence, attenuation in direction of propagation.

**Funding:** This research received no external funding.

**Institutional Review Board Statement:** Not applicable.

**Informed Consent Statement:** Not applicable.

**Data Availability Statement:** Not applicable.

**Acknowledgments:** Not applicable.

**Conflicts of Interest:** The authors declare no conflict of interest.

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
