# Peer review of "Appearance of Supersonic Stoneley Waves in Auxetics"

_crystals, doi:10.3390/cryst12030430_

Round 1

Reviewer 1 Report

The authors study Stoneley waves in auxetic material, and derive conditions for density and Poisson's ratio of auxetic materials for supersonic waves to propagate. The paper is of clear interest to the reader of the journal; it needs however a more detailed level of explanations for the reader's understanding.

  • The notions of Stoneley waves needs a better explanation: what are these waves, their characteristics?
  • The originality of the proposed work needs to be better underlined in comparison to literature works.
  • Condition (6) needs to be explained: what does it represent and how is it obtained
  • The notion of supersonic waves needs to be explained.

Author Response

The author thanks the respectable Reviewer for valuable comments, all of which were addressed in the revised version of the manuscript

  • The notions of Stoneley waves needs a better explanation: what are these waves, their characteristics? The definition for Stoneley waves and the most important characteristics are given in the Introduction.
  • The originality of the proposed work needs to be better underlined in comparison to literature works. The originality is emphasized and compared with other works on Stoneley wave propagation, some more references is added.
  • Condition (6) needs to be explained: what does it represent and how is it obtained. The condition (6) is explained.
  • The notion of supersonic waves needs to be explained. The definition for the supersonic wave is given.

Reviewer 2 Report

The manuscript crystals-1629377 covers the topic that is of relevance and interest to Crystals journal. It presents an analytical expression for the supersonic Stoneley waves in function of the negative Poisson’s ration and young modulus of the contacting isotropic media. Two fundamental points need clarification:

  1.     The authors should highlight the main originality of the work. The Wiechert condition and the secular equation of the Stoneley waves are presented in the literature;

Kuznetsov, Sergey V.. “Stoneley waves in auxetics and non-auxetics: Wiechert case.” Mechanics of Advanced Materials and Structures 29 (2020): 873 - 878.

  1.     It is well known that the poisson’s ratio for isotropic material will varies between 0.5 and 1. We can observe in Fig.1 where the author presents the variation of Poisson’s ration in function of the effective density, that Poisson’s ratio varies from 0 to -1 which cannot be the case of isotropic material.

  1. Many keywords were used without explication.

  1. Many relevant paper in the literature in the same field should be cited:
  2. Reda, H., Elnady, K., Ganghoffer, JF., Lakiss, H. Nonlinear wave propagation analysis in hyperelastic 1D microstructured materials constructed by homogenization. Mechanics Research Communications, 84, 136-141, 2017

I think that these are minor issues and the manuscript is very poor of some definition of parameters and physical meaning, the authors need to clarify some points before published in Crystals.

Author Response

The author thanks the respectable Reviewer for valuable comments, all of which were addressed in the revised version of the manuscript.

1) The authors should highlight the main originality of the work. The Wiechert condition and the secular equation of the Stoneley waves are presented in the literature; Kuznetsov, S.V.. “Stoneley waves in auxetics and non-auxetics: Wiechert case.” Mechanics of Advanced Materials and Structures 29 (2020): 873 - 878. This paper is cited and the key differences between current work and the cited paper are presented. 

2) It is well known that the Poisson’s ratio for isotropic material will varies between 0.5 and 1. We can observe in Fig.1 where the author presents the variation of Poisson’s ration in function of the effective density, that Poisson’s ratio varies from 0 to -1 which cannot be the case of isotropic material. The author thanks the Reviewer, the corresponding explanation for isotropic materials with negative Poisson's ratio (the so called auxetics) is given.

3) Many keywords were used without explication. The corresponding explications are given.

4) 

Many relevant paper in the literature in the same field should be cited:

    Reda, H., Elnady, K., Ganghoffer, JF., Lakiss, H. Nonlinear wave propagation analysis in hyperelastic 1D microstructured materials constructed by homogenization. Mechanics Research Communications, 84, 136-141, 2017. Many thanks, this work is cited.